# The Influence of Hydroponic Potato Plant Cultivation on Selected Properties of Starch Isolated from Its Tubers

**DOI:** 10.3390/molecules27030856

**Published:** 2022-01-27

**Authors:** Marta Liszka-Skoczylas, Wiktor Berski, Mariusz Witczak, Łukasz Skoczylas, Iwona Kowalska, Sylwester Smoleń, Paweł Szlachcic, Marcin Kozieł

**Affiliations:** 1Department of Engineering and Machinery for Food Industry, Faculty of Food Technology, University of Agriculture in Krakow, al. Mickiewicza 21, 31-120 Krakow, Poland; rrwitcza@cyf-kr.edu.pl; 2Department of Carbohydrates Technology and Cereals Processing, Faculty of Food Technology, University of Agriculture in Krakow, al. Mickiewicza 21, 31-120 Krakow, Poland; wiktor.berski@urk.edu.pl; 3Department of Plant Product Technology and Nutrition Hygiene, Faculty of Food Technology, University of Agriculture in Krakow, al. Mickiewicza 21, 31-120 Krakow, Poland; lukasz.skoczylas@urk.edu.pl; 4Department of Plant Biology and Biotechnology, Faculty of Biotechnology and Horticulture, University of Agriculture in Krakow, al. Mickiewicza 21, 31-120 Krakow, Poland; iwona.kowalska@urk.edu.pl (I.K.); sylwester.smolen@urk.edu.pl (S.S.); 5Department of Chemistry, Faculty of Food Technology, University of Agriculture in Krakow, al. Mickiewicza 21, 31-120 Krakow, Poland; pawel.szlachcic@urk.edu.pl; 6Department of Crystal Chemistry and Crystal Physics, Faculty of Chemistry, Jagiellonian University, Gronostajowa 2 Str., 30-387 Krakow, Poland; marcin.koziel@uj.edu.pl

**Keywords:** potato starch, pasting properties, differential scanning calorimetry, wide-angle X-ray scattering, hydroponics

## Abstract

Starch is a natural polysaccharide for which the technological quality depends on the genetic basis of the plant and the environmental conditions of the cultivation. Growing plants under cover without soil has many advantages for controlling the above-mentioned conditions. The present research focuses on determining the effect of under cover hydroponic potato cultivation on the physicochemical properties of accumulated potato starch (PS). The plants were grown in the hydroponic system, with (greenhouse, GH) and without recirculation nutrient solution (foil tunnel, FT). The reference sample was PS isolated from plants grown in a tunnel in containers filled with mineral soil (SO). The influence of the cultivation method on the elemental composition of the starch molecules was noted. The cultivation method also influenced the protein and amylose content of the PS. Considering the chromatic parameters, PS-GH and PS-FT were brighter and whiter, with a tinge of blue, than PS-SO. PS-SO was also characterized by the largest average diameters of granules, while PS-GH had the lowest crystallinity. PS-SO showed a better resistance to the combined action of elevated temperature and shear force. There was a slight variation in the gelatinization temperature values. Additionally, significant differences for enthalpy and the retrogradation ratio were observed. The cultivation method did not influence the glass transition and melting.

## 1. Introduction

Starch is the main reserve material of the higher plants and at the same time it is the most important carbohydrate in the human diet. The constant, high demand for starch is a result of its versatile applications in various industries, especially in the food industry. Starch greatly imparts textural properties to many food products and has industrial applications as a thickener, stabilizer, gelling agent, filler, water-retaining agent and even as an adhesive. The physicochemical and functional properties of aqueous starch systems and their uniqueness in various applications, differ depending on the botanical origin of this biopolymer. The main sources of starch for industrial production are maize (normal and waxy starch), wheat, potato and cassava [1].

In terms of technology, potato starch (PS) is considered the best, which is related to its physicochemical properties. This uniqueness results from the large size of its granules, relatively long chains of amylose (AM) and amylopectin (AP), the presence of ester-bound phosphate groups in the AP molecule, the ability to exchange some cations with a corresponding effect on the viscosity and the ability to form thick viscoelastic gels after heating and then cooling [2,3]. This is due to the presence of phosphate esters and a low lipid content [4,5].

However, there are significant differences between the physicochemical properties of potato starch obtained from different potato tuber varieties [6]. Many publications on potato starches assume that variety influences the composition, structure, and physical properties of a molecules [7,8,9,10,11,12,13,14]. However, there is also evidence [6,15,16] that factors beyond the genetic factors are equally important. These include the environmental factors and agrotechnical methods applied during plant cultivation. The influence of these factors on the structure and properties of starch is examined in the present paper, in terms of the planting date; harvest time (potato development stage); length of growing season; temperature during plant growth; day length; type and intensity of light; amount of precipitation; carbon dioxide concentration; soil composition; mineral and organic fertilization and tuber storage conditions (temperature, humidity and time) [6,15,16,17,18,19,20,21].

In many cases, these influences are greater than the varietal differences or even the differences between the species, which are reflected in the quality of the raw material and as a consequence in the quality of the product. These factors are difficult to control in field conditions due to the unpredictable weather conditions [22]. The influence of the environmental conditions on the physicochemical properties of starch can be more easily determined and consequently better understood in experiments conducted in fully controlled conditions (cultivation under cover).

The use of soilless methods of growing vegetables in foil tunnels and greenhouses has become more and more popular. Compared to the conventional cultivation in soil, hydroponic cultivation ensures the faster growth and development of plants resulting in higher yields. Plants from such cultivations are of a high quality with good post-harvest durability. In addition, they are characterized by a higher health, because due to the specificity of the cultivation, they are free from pests and soil borne diseases [23].

Little information has been published so far on potato growth in soilless systems, although the research indicates that hydroponic systems, especially the Nutrient Film Technique (NFT) and aeroponics, have a potential commercial applications in the potato industry for certified seed production [24].

The literature on the subject contains several data documenting the possibility of growing potatoes in hydroponic systems [25,26]. They are mainly aimed at determining the possibility of cultivating this species in such systems. Only microtubers (weighing a few grams) were obtained, not regular size tubers [25,26,27,28]. The exception was the study conducted by Wheeler et al. [24], who achieved a much higher yield and size of potato tubers when growing in a hydroponic system. Smoleń et al. [29] also obtained tubers at the full physiological maturity stage in their research on the enrichment of potato tubers with mineral nutrients, by applying the NFT technique.

An important aspect of hydroponic potato cultivation is its use for the cultivation/selection of new varieties and, e.g., for biogenerative life support systems [30].

Due to the fact that the physicochemical properties of starch, which constitutes about 17–21% of the fresh potato tubers mass, determine the quality and direction of the technological application of both the tubers and the biopolymer itself, an attempt is made to determine the impact of the hydroponic cultivation conditions of potato plants, and on the chemical composition and physical properties of starch isolated from such tubers.

## 2. Results and Discussion

### 2.1. Composition Analysis

The starch mineral composition can be of decisive importance for its industrial application, because the degree of phosphorylation determines the physicochemical properties of the polymer [31,32,33]. Moreover, it is likely that the level of divalent cations, such as calcium and magnesium, has a significant influence on starch gelatinization properties through the ionic cross-linking of phosphate starch esters [34,35,36]. The investigated starch samples were characterized by a low ash content (Table 1). The higher values, compared to the SO sample (control; two-year average 0.20%), were recorded for both PS-GH and FT (two-year average by 40% and 65%, respectively). All the obtained values were within the range reported for the PS [15,37,38,39] and within the limit of 0.5%, recommended for industrial grade A starches [40].

The non-carbohydrate compound found in relatively high amounts in the PS was phosphorus. Its high content in the PS is desirable in order to obtain a high viscosity paste [41]. The content of phosphorus in the analyzed PS (Table 1) ranged from 540.25 mg/kg for SO-II (second year of cultivation) up to 745.26 mg/kg for FT-II. Taking into account the two-year average content of P in PS, it can be concluded that the hydroponic cultivation of plants improves the accumulation of this element in starch. In the soil environment, the phosphorus uptake by plants was limited by the chemical sorption of this element. Chemical sorption of phosphorus takes place mainly with iron, aluminum and manganese compounds. Thus, for any plant, phosphorus uptake from the soil is much more difficult than from the nutrients in hydroponics. Therefore, the phosphorus content of starch from hydroponic potatoes was higher than in soil. We also observed a statistically important difference in the content of this macronutrient in the polymer between different hydroponic systems (closed vs. open one). The phosphorus content in GH (closed system) was on average 6.28% lower than in FT (open system). The presented values are comparable with the values given in other publications [6,21,42].

The PS contained natural metal cations bound by ionic forces to the phosphate ester groups. The level of divalent cations, such as calcium and magnesium, has a significant influence on the pasting properties of starch, possibly by ionically cross-linking the starch phosphate esters [34,35,36,43]. De Willigen et al. [44] reported that the characteristics of ionically cross-linked starch paste should be similar to those of covalently cross-linked starch.

The content of calcium in potato starch varied widely, from 21.89 to 98.41 mg/kg (Table 1). These values are consistent with those reported by other researchers [16,36,38,45]. The average two-year content of calcium ions in the PS extracted from plants grown in both NFT systems is significantly lower than for reference sample (SO) (for GH by 77%, whereas for FT by 71%). Such a large difference, although statistically significant, was not recorded between GH and FT.

The content of magnesium ions in the studied starches ranges from 46.26 to 73.54 mg/kg (Table 1), which is consistent with the literature data [36,38]. Additionally, for this element, the average two-year content in the PS isolated from the plants grown in the soilless system is significantly lower than for SO (for GH by 34.5%, and by 31.2% for FT). A slight difference of 2.37 mg/kg was observed between the open and closed NFT systems.

The content of the monovalent metal ions (K^+^, Na^+^) ranged from 524.35 to 920.05 mg/kg and from 6.62 to 25.02 mg/kg, respectively (Table 1). The data related to the content of these elements in the PS were already published [34,36,38]. Additionally, in the case of the content of the minerals mentioned above, a difference was observed between the PS isolated from the plants grown in soil (SO) and by the hydroponics method. The K^+^ content in SO sample was lower on average by approximately 37.5% than in other samples. On other hand, for the Na^+^ content, a reverse situation was observed—for the GH sample, it was over two times lower, whereas and for the FT sample, it was one and a half times lower than for the SO. In the case of these ions, the influence of the NFT system (open/closed) on their accumulation in the starch granules was also visible. In the closed system (GH), the content of potassium ions was higher, and in the case of the sodium ions, it was lower than in the open system without nutrient recirculation (FT). According to the literature data [14,46,47], the cations content was relatively higher in the PS with an elevated phosphorus content. The results of this work seem to confirm these observations, because the sum of the content of calcium, magnesium, potassium and sodium in the analyzed starches is 991.15, 979.47 and 756.00 mg/kg for GH, FT and SO, respectively, while a significantly higher P content is noted for the starch isolated from potato tubers from hydroponic cultivation (GH and FT). Sodium is an element commonly found in soil, but this element is not introduced into the nutrients in hydroponic cultivation. Therefore, the sodium content in starch isolated from the potato tubers grown in soil is much higher than that of the starch from the NFT crops. In addition, there is a competition (antagonism) between the potassium ion (K^+^) and the sodium ion (Na^+^) in plants during root uptake. Since the uptake of potassium from nutrients is much easier than from soil, the K versus Na interaction in soil is weaker than in nutrients. Divalent ions and potassium are antagonistic. The high content and easy uptake of potassium in hydroponics weakens the uptake of Ca^2+^ and Mg^2+^. These conclusions also confirm the results obtained for the molar fraction of mineral elements (Table 1).

An important characteristic of starch is the amylose (AM) content. Due to the very low lipid content of potato starches, the observed AM content values are equivalent to both the amount of apparent amylose and the amount of total amylose. In the PS samples tested, the AM content ranged from 26.18% to 27.82% (Table 1), which is consistent with values reported by other researchers [14,15,33,38,48,49,50]. The lowest AM content, among the studied samples, was characterized by FT starch (two-year average 26.34 g/100 g). The values higher by 0.70 g/100 g and 1.48 g/100 g were observed for SO and GH samples, respectively, which correspond to an increase in the AM content by 2.66% and 5.62%, compared to FT. The ratio of amylose to amylopectin (AP) ranges from 1:2.59 (GH) to 1:2.80 (FT), and reflects the distribution (spread) of the AP content [6]. Our results confirm that the ratio of AM to AP is a relatively constant characteristic of the plant botanical variety, but can be modified by cultivation [14]. The range of amylose content that we observed (SD/mean = 0.027 for all starch samples from the potato variety ‘Vineta’) was as expected.

The low protein (and also ash) content of the investigated starches, ranging from 0.26 g/100 g to 0.33 g/100 g (Table 1), indicates its purity and is consistent with the data reported by other authors [6,48]. The lowest protein content (two-year average nitrogen content) of 0.043g N/100g was recorded for the SO sample. The starch obtained from tubers grown under the NFT system contained statistically more of it (0.046g N/100 g and 0.053 g N/100 g for GH and FT, respectively).

### 2.2. Starch Color

Color is one of the most important visual quality parameters for both raw materials and products prepared on their basis. The color of the raw material can provide information on the chemical composition of the product, and thus determine its suitability for processing. Moreover, its variation confirms changes that occur in the structure or chemical composition of the raw material or product during processing or storage.

The results of the color measurements of the tested potato starch samples are presented in Table 2. The color parameter values obtained are similar to those presented in the literature [51,52]. Regarding the lightness (L*), it intensifies for all the starches isolated from potato tubers from the NFT crop; in relation to the starch from the SO object, the highest L* value is marked by the GH starch (two-year average L* = 92.05), while the SO starch has an L* value 8.43 lower and the FT starch by 2.69 (Table 2).

The change in the plant cultivation system from soil to soilless is significantly reflected in the color a* (greenness to redness) and b* (blueness to yellowness) parameters (Table 2). For the starch isolated from plants from the NFT cultivation, the a* parameter takes negative values, indicating a shift in color towards greenness, whereas for the SO sample, the color shifts towards red (a* > 0). The b* parameter for all the tested starches obtained positive values; however, for the GH and FT starches (b* 2.7), it shifted more towards blue than for the SO starch (b* = 6.06).

From the color parameters L*, a* and b*, the total different color ΔE (Table 2) was calculated, which reflects the color difference between the two compared samples. The values of ΔE 2.3 corresponds to a just noticeable difference [53]. When considering the effect of the cultivation method on this parameter, the lowest ΔE values, and therefore the least color difference, were recorded between the hydroponic cultivated starches (Table 2). The ΔE values obtained for GH in relation to FT (1st year of cultivation) indicated that there was no difference in color between the studied samples that could be noticed by the observer (ΔE = 0.9). There is a noticeable color difference between the other samples and this is most clearly evident in the comparison of the PS from hydroponics to SO. It should also be noted that the year of cultivation is important for this parameter.

The hydroponic cultivation also causes a noticeable decrease in the saturation of the potato starch color, as evidenced by the chroma C* value (Table 2). Additionally, the whiteness index (WI) confirms that whiter starch was obtained from objects where the soilless cultivation of plants was applied (WI increase for SO ≈ 100%, compared to GH and FT) (Table 2).

It should be emphasized that all the aspects related to the chromatic attributes of the tested starches confirmed the influence of the type of applied potato plant cultivation method—its change from soil to hydroponic. The starch samples obtained from the tubers grown in soil are darker and grayer with red and yellow tones, compared to the other samples, which can be concluded from the lower values of the L* parameter and higher values of the a*, b*, C* and WI indices.

### 2.3. Particle Size Distribution (PSD) of the Potato Starch

Potato starch exhibits a unimodal PSD, and among the granules can be observed a fraction of small-spherical granules and large-ellipsoidal ones. Since the structure and size of the starch granules can influence the physical properties of starch [54,55,56], it is therefore crucial to determine the PSD of the investigated PS (Table 2). The average diameter of the potato starch granules over was 20 µm, with the majority of the granules below 50 µm. The potatoes grown in soil (SO) were able to synthetize the granules with the highest average diameter, whereas, in the case of the FT starch, it was the lowest. The obtained average diameters of PS granules were consistent with the data presented by other authors: 23.0–30.9 µm [42].

A significant correlation was observed between the content of the large granule fraction (>70 µm) and the content of calcium, magnesium and sodium (positive) and potassium and phosphorus (negative). The last correlation that related the phosphorus content vs. large granules size content, was also observed by Sikora et al. [57].

### 2.4. Water Binding Capacity (WBC), Welling Power (SP) and Starch Solubility (S)

The temperature dependence of the WBC, SP and S parameters obtained for the analyzed PS samples is presented in Table 3.

The WBC was influenced by both the temperature and cultivation method. At each measurement temperature, SO had the highest WBC values. The increment of the WBC values between the lowest and the highest measurement temperature was 74.14 and 68.26 g/100 g for GH and FT, respectively, compared to 75.81 g/100 g for SO. The increase in the WBC values with the increasing temperature can be due to gelatinization, which breaks the weak associative bonds in the amorphous region of the starch granules and allows for increased hydration [58]. The WBC values obtained were slightly lower than those reported in the literature (77.2–89.0%) [11,59].

The SP indicates the interaction between the amorphous and crystalline region of the starch granules. At the pasting temperature, the starch granules have limited swelling properties and therefore only a small amount of starch is dissolved, but at a higher temperature there is an increase in the SP value and a large amount of starch leaks out [60]. The highest increase in the swelling power (7.5 to 8 times) was observed when the temperature was increased from 60 to 70 °C, with the most pronounced increase for the GH sample (Table 3). At the highest temperature (90 °C), the FT (two-year average 92.70%), GH (two-year average 100.46%) and SO (two-year average 105.15%) starches showed large differences in the SP (*p* < 0.05). At 80 °C, the differences in the SP of the tested starch samples was relatively low, but still it was statistically significant. At the other temperatures, the swelling power of the SO was higher than that of the other samples.

By analyzing the obtained results, it was noted that the calcium content was positively correlated with the swelling power at all temperatures, which can be explained by the across-linking of starch chains by covalent bounds created by this divalent ion. A similar observation was made by other authors [61] (therefore, the calcium content contributes with the water diffusion to the coarse particles and improves the swelling of the starch granules).

Solubility (S) is the measure of the amount of solutes that were washed out of the starch granules when measuring their swelling ability. The starch solubility increased with the increasing temperature. At high temperatures, an increase in the S value indicates an increase in the amount of solute amylopectin, the amount of which increases dramatically when the granules are ruptured [62]. The solubility of starch was significantly affected by both the temperature and the method of potato plant cultivation (soil/soilless system) (Table 3). An increase in the solubility with an increasing temperature was observed in every starch sample tested, with the most intense increase between 60 and 70 °C (temperature close to the pasting temperature—Table 4). SO solubility increased by 5.6 times, while for the PS samples from both types of hydroponic systems the increases were not as pronounced (3.6 and 4.7 times for GH and FT, respectively). Between the temperatures of 70 and 80 °C, the intensity of S increase was not so significant. In the range of these temperatures, the smallest increase was recorded for the SO (about 1.5), and about 2.7 times for the remaining samples. The solubility of SO was 24.14 g/100 g and for the remaining samples these values were lower by 1.4 and 1.9 g/100 g for GH and FT, respectively.

For all the investigated parameters (with few exceptions, namely WBC at 80 °C and S at 60 °C), a statistical difference was recorded between the I and II cultivation year (Table 3).

### 2.5. Pasting Properties

The results of the pasting properties of 5% potato starch suspensions in water are presented in Table 4 and Figure 1. From observing the pasting curves, it can be concluded that three distinctive courses can be observed for the starches grown in different conditions (GH, FT and SO), with the largest discrepancies observed between the cultivation year for the SO starches. The pasting temperatures (PTs) of all the investigated PS was within a rather narrow range, 69.5–72.1 °C, which are in the range reported in the literature [63,64] and also similar for oat starches [65].

This parameter was affected by the cultivation method, but not by the year of cultivation. The peak viscosity (PV) was varied significantly among the samples, with the lowest values attained by samples grown in soil (SO average 612 BU) and the highest values were attained by the samples grown in the foil tunnel (FT, 1253 BU), clearly indicating the importance of the growing method. Additionally, the differences were spotted for the PVt (time needed to reach PV) and PVT (temperature at PV) parameters, and the lowest values were observed for FT starches, which were characterized by a sharp increase in viscosity followed by a dramatic decrease in it (high values of BD and BD%). Due to the presence of a covalently bound phosphorus moiety [41], the PS can reach high maximum viscosity values, much higher than for cereal starches [63,64,66], where phosphorus is incorporated in the form of phospholipids restricting granular swelling, and thus limiting maximum viscosity.

In this research, that a well-known correlation between a phosphorus content and starch paste viscosity was observed [32,41,48,54]. The obtained PV values for SO starches were lower than for other cultivation methods; in fact, their PV values were similar to that observed for cereal starches [65].

As previously mentioned, the pasting of FT starches was characterized by a rapid growth of viscosity at the first stage of this phenomenon, which was reflected by the appropriate parameters of the logistic model [67], namely the highest values of V_peak_ and s, and the lowest r V_peak_ can be identified with PV, but they were generally slightly lower than the corresponding PV values, with the exception for SO I and II starches. Another parameter applied in this model, r, was the time needed to reach half of the V_peak_ value, and the highest values were observed for SO starches, indicating a rather slow swelling of the granules, whereas the lowest values were calculated for the rapid viscosity growth observed for the FT samples.

The last, dimensionless parameter of the model (s) is related to granular swelling. As can be observed for slowly developing viscosity SO starches, the value is almost half that of the GH, and over two times lower than for the FT starches.

Starches subjected to the combined action of elevated temperature and shear forces, decreased their viscosity during the holding period, which was manifested by appropriate MV (minimum viscosity), BD (breakdown), BD% and HPSI (hot paste stability index) values. This drop in viscosity was related to the disintegration of swollen starch granules, and can describe the resistance of swollen starch granules against extreme conditions. It was particularly noticeable for FT starches, where a dramatic decrease in viscosity was observed. On the other hand, other starches were characterized by a rather low viscosity drop (small disintegration of swollen starch granules), and by the relatively high stability of viscosity during holding at elevated temperatures (high values of HPSI and low BD and BD% Table 4). A convenient method to compare this decrease in viscosity for different starches (or for analyses performed using different procedures or equipment) is the BD% value (and at the cooling stage, an increase in the viscosity could be described by means of SB%). The BD% values reported for the potato starches varied in a broad range, from 11.7 to 58.3 [63,64], indicating different responses on shearing at elevated temperatures. The calculated BD% values for SO starches (8.8%) in this research were below the lower limit of this range, indicating good starch paste resistance (that was also confirmed by high HPSI values), whereas others (FT and GH) were within it. For the starches of a different botanical origin, the BD% were as follows: waxy maize (50.5–73.1%), maize (12.8–43.7%), wheat (20.3–32.0%), oat (35.9–36.8%), tapioca (71.2%) and pea (59.3) [65,67,68,69].

In order to describe hot starch paste stability, HPSI can also be applied (Table 4). The values calculated in this research were found within a broad range (64.6–96.3%), with the highest value observed for SO starches indicating their good stability. The HPSI values for waxy maize starch were 89.0% and 76.4% for regular maize [69].

At the end of holding, the MV value was observed (from 492 to 819 BU) and then, as a result of cooling (and the creation of hydrogen bonds), an increase in viscosity was observed of up to TV (from 848 to 1375 BU). This increase in viscosity is described as a setback (SB), and could be related to the starch retrogradation. The SB% values obtained in this research varied from 36.4 (SO-I) to 48.1% (FT-I). The calculated SB% (based on literature data) for the potato starches varied from 11.8 to 24.5 [63,64].

### 2.6. Thermal Properties

The functional properties of starch make it one of the most widely used raw materials in the food industry. Its technological potential can be further enhanced by various modifications or the addition of nutrients, which affect the physical and functional properties of a given product. Among the many characteristic properties of starch, one of the most important is the phenomenon of pasting and the subsequent retrogradation of starch. Starch pasting occurs during many food processing operations and has a significant impact on the properties of the resulting products. Starch granules, as a result of water absorption at elevated temperatures, swell and disintegrate, releasing AM, which is poorly soluble in water, and AP. The pasting temperature and the process itself depend mainly on starch type and the amount of available water. The obtained starch pastes and gels are subjected to the retrogradation process. The effect of this phenomenon is the transition of biopolymers from a form of paste into a partially ordered form, and the formation of a crystalline network. During this process, water molecules are expelled from the biopolymer network as a result of the reduction in intermolecular spaces. This leads to structural changes during storage, resulting in the increased turbidity of pastes and gels, increased gel stiffness, syneresis, as well as bread staling.

The results of the thermodynamic analysis of starch gelatinization and retrogradation are presented in Table 5. Small differences in the gelatinization temperatures were observed. Only in the case of T_Pg_ and T_Eg_ did the samples slightly differ, which indicated a different course of the gelatinization process, probably related to a significant difference in the SP values, especially at higher temperatures. The values of all the temperatures analyzed were relatively high and were at or above the upper limits of the values reported in the literature for PS [33,63]. The values of gelatinization enthalpy ranged from 15.59 J/g to 17.2 J/g and were not significantly different from a statistical perspective. These values were within those reported in the literature for PS [33,63].

As the starch paste cools, AM forms double helixes composed of 40–70 glucose molecules and AP forms crystalline structures [33,63]. This process, referred to as starch retrogradation, is associated with the formation of hydrogen bonds between starch chains [33].

The values (temperatures, enthalpies) obtained with DSC (Table 5) characterizing this process were generally lower than those characterizing the gelatinization process. In the case studied, the values of the temperature of the onset of transformation (T_Or_) ranged from 44.6 °C to 50.13 °C and depended on the method of cultivation. The two samples from hydroponic cultivation (GH and FT) did not differ from each other. The peak temperature (T_Pr_) value did not differ, while a similar relationship was observed for the end of the transformation (T_Er_) as for the beginning (T_Or_), but the variation was slightly smaller. Significant differences were found for the enthalpy of retrogradation.

Samples from conventional cultivation (SO) were not different from the GH sample, while both were significantly different from the FT one. A similar relationship was characterized by the degree of retrogradation (R%). This indicated, on the one hand, the similarity between traditional and hydroponic cultivation, and on the other hand, it indicated the significant effect of the recirculation of the nutrient solution on the PS properties and its behavior during the retrogradation process. The stability of starch and its products strongly depends on their composition and parameters characterizing the storage site (relative humidity and temperature) [70,71]. When determining the appropriate storage conditions, the vitreous transition temperature (T_g_), a property extremely important from the point of view of the transformations occurring in food, plays a crucial role. It characterizes the change in mobility of the water contained in food products and is related to the interactions between the water molecules and macromolecules of other food components. The values of the temperatures characterizing the phenomenon of vitreous transition and the subsequent melting peak are summarized in Table 5. In this case, no statistically significant differentiation of samples was found. Only in the case of the onset of vitreous transition and the enthalpy of melting, slightly lower values were found for the conventional cultivation (SO), than for the other two samples derived from hydroponic cultivation (GH and FT).

### 2.7. Crystallinity

The crystallinity of starch granules can reveal important information about the internal structure and the type of amylose chains distribution within the granules. There are two distinct crystalline polymorphic forms: the A-type (B2 space group) [72], mostly found in cereal starches, and the B-type (P6_1_ space group) [73,74], observed in tuber starches (PS is in that group) and high-amylose cereal starches. As starch is composed from mostly amorphous amylose and semicrystalline amylopectin, the powder XRD pattern consists of amorphous part and crystalline peaks (compare Figures 1 and 3 in [75]). In our case, the XRD patterns of the analyzed starches are presented in Figure 2a. As can be seen from Figure 2a, in all the cases, the shape of the curves is similar. The crystalline diffraction peaks can be seen in all the cases, and the best patterns are observed for GH I and GH II starches. Figure 2b shows the XRD pattern for GH II after data processing (background and amorphic part subtraction). In Figure 2b, one can recognize crystalline peaks, the most important are 5.74, 10.10–11.74, 14.38, 15.18, 17.26, 19.86, 22.40, 24.22 and 26.36°. Most of them indicate a B-type structure, but a peak at 22.40° could also indicate some fraction of the C-type [75,76]. The relative crystallinity of the samples was calculated by comparing the crystalline area with total area of the peaks in the 2θ range 4–30°, using the Equation (2) from the paper of Frost and co-workers [77]. The results are summarized in Table 6. As can be seen from the table, the highest values are found for the starches obtained from the potatoes cultivated in a greenhouse (GH I and II), which is consistent with the highest amylose content (compare Table 1). The other samples have a significantly lower crystallinity.

## 3. Materials and Methods

### 3.1. Plant Material

Potato starch (PS) was extracted from the tubers of potato (*Solanum tuberosum* L.) cv. ‘Vineta’. The plants were grown in two-year experiments in the spring season under cover at the Faculty of Biotechnology and Horticulture at the University of Agriculture in Krakow. The cultivation was carried out in three systems. The first one included hydroponic cultivation using the Nutrient Film Technique (NFT) system, with a nutrient solution recirculation (closed system) in a greenhouse (GH); the second was the hydroponic cultivation of NFT without the recirculation of the nutrient solution (open system) in a foil tunnel (FT); and the third (as reference sample) was cultivation in containers filled with mineral soil (SO) in a foil tunnel.

In each of the three cultivation systems, certified seed potatoes of class C/B (grown from the breeder’s basic material) of the caliber of 35–55 mm were applied. Both NFT experiments consisted of 3 replications with 10 plants in each replicate.

The details of hydroponic potato cultivation in a greenhouse are presented in our earlier publication [29]. The potato seeds were planted in NFT beds lined with a fiber water-ascension matt. The surfaces of troughs were covered with two-sided white-and-black foil. No additional substrate was used. The greenhouse was equipped with NFT set with a 1300 dm^3^ medium container, facilitating potato cultivation in recirculation hydroponics.

In a foil tunnel, potato plants were grown in gutters filled with perlite. The supply of water and nutrients to the cultivation gutters was carried out with the help of the nutrient solution administered with the drip system. The frequency of nutrient solution feeding was regulated on the basis of the observation of moisture in the subsoil mats.

In both hydroponic experiments, the composition of the nutrient solution was adjusted to the growth phase of the plants. Two types of nutrient solution were applied: the first one in the phase of vegetative growth, and the second one in the period of flowering and tuberization. The composition of both nutrients was presented elsewhere [29,78]. In both cultivation systems (GH, FT), the pH of the nutrient solution was kept in the range of 5.5–6.0. The day and night temperatures were set to 22 °C and 17 °C, respectively. Potato tubers were collected in the phase of yellowing and drying up of the plant leaves and stems.

The soil cultivation experiment (SO) consisted of 5 replications (30 plant containers). Containers with dimensions of 60 × 40 × 20 cm were filled with mineral soil classified as light dusty clay with a grain size composition: 35% sand, 28% dust and 37% clay, with an average content of organic matter at the level of 2.76%. According to the method [79] most commonly used in potato fertilization practice, the number of seed potatoes per one container was 2. The containers were in the same foil tunnel as the hydroponic cultivation (FT). The plants in the containers were watered with the same amount of tap water. The collection was carried out at the stage of the full maturity of the plants.

### 3.2. Starch Isolation

The method described by Singh and Singh [80], with slight modifications, was applied for the starch isolation. After the deformed potato tubers were separated, the rest of the tubers were thoroughly cleaned, peeled and washed in cold distilled water. The peeled tubers (1 kg) were cut into pieces with a knife and then homogenized with the addition of distilled water using a blender equipped with razor blades. The resulting suspension was filtered through two layers of gauze. In order to wash the starch granules from the potato remnants, the suspension on the gaze was washed several times with an excess of cold distilled water. The starch granules suspended in the filtrate were left overnight. The decanted starch cake was suspended in distilled water, allowed to settle and then decanted. The process was repeated 5 times. The washed starch was dried in an oven (40 °C, 24 h), then ground in a mortar and sieved through a 0.250 µm mesh sieve. Prepared in such a manner, the starch was stored in glass air tight containers until further examination/investigation [59,80].

### 3.3. Physicochemical Properties

Dry matter (g/100 g) and the ash content (g/100 g) were determined according to the AOAC method [81].

The starch apparent amylose content (g/100 g) was determined using the iodine spectrophotometric method described by Morrison and Laignelet [82] using a UV/VIS spectrophotometer (Shimadzu UV-160 A, Kyoto, Japan).

The total nitrogen and protein content (N × 6.25) were determined by the Kjeldahl method according to the AOAC standard [81]. Digestion Unit K-424 (Büchi, Uster, Switzerland) was used for the mineralization of the samples, and Distillation Unit B-324 (Büchi, Uster, Switzerland) for distillation.

The total content of phosphorus, calcium, magnesium, potassium and sodium was determined by emission spectrophotometry using the ICP-OES Prodigy spectrophotometer (Teledyne Leeman Labs, Hudson, NH, USA), after microwave mineralization of the samples in 65% super pure HNO_3_ [83]. For Ca, Mg, K and Na, a molar fraction was calculated. The color characterization of starch was performed by the transmittance method in the CIE Lab system [84]. The L*, a* and b* parameters were determined by reflection method on a MINOLTA CM-3500d (Konica Minolta, New York, NY, USA) with Illuminant D65 as standard by direct 3-fold surface measurement of the starch film placed on the transparent Petri dish (6 cm diameter and 4 cm high) at different locations. Measurements were made using a 30 mm diameter membrane and a with a measurement angle of 10°. The measurements were allowed for the determination of the following parameters:

L*—lightness (L* = 0 for black, L* = 100 for white);

a*—a color parameter ranging from green (−a*) to red (a*);

b*—a color parameter ranging from blue (−b*) to yellow (b*).

The quantitative color parameter is the chroma C*, determined from the equation below (Equation (1)):(1)$$C^{*}=\sqrt{a^{*2}{+b}^{*2}}$$

The color difference between starch from the hydroponic cultivations (GH and FT) and starch from soil cultivation (SO) was expressed as the total color difference ΔE*ab. This parameter was computed as the Euclidean distance between two points in the three-dimensional space defined by L*, a* and b* using the following equation (Equation (2)):(2)$$\Delta E_{\mathrm{ab}}^{*}=\sqrt{\Delta a^{*2}+\Delta b^{*2}+\Delta L^{*2}}$$
where Δa*, Δb*, ΔL* is the difference between the value of the respective parameters for starch from SO vs. GH and FT.

Whiteness index is a parameter linking together all the previously mentioned parameters into a single indicator. According to Rhim et al. [85], it indicates the whiteness degree, and is calculated as (Equation (3)):(3)$$\mathrm{WI}=\sqrt{\left({100\;-\;L*}^{2}\right){+a*}^{2}{+b*}^{2}}$$

The particle size distribution (PSD) of the potato starch granules was calculated by means of video enhanced microscopy (VEM) applying image analysis [86].

### 3.4. Water Binding Capacity (WBC), Swelling Power (SP), Starch Solubility (S)

Water binding capacity, swelling power and starch solubility in water at 60, 70, 80 and 90 °C was measured according to Richter et al. [87]. Starch dispersion (1 g dry basis (db) in 80 cm^3^ of distilled water) was heated at the indicated temperature in a water bath during 1 h with constant agitation, followed by a rapid cooling to room temperature and centrifugation at 2103× *g* for 15 min. The formed precipitate was weighed and calculated as the SP value. The supernatant was dried on a dish, weighed and the S was determined.

### 3.5. Pasting Properties

Starch pasting properties were performed as previously described by Berski et al. [65], using a Micro Visco Amylo-Graph (Brabender, Duisburg, Germany). A total of 5% (*w*/*w*) of water-based starch dispersion was pasted using the following temperature program: the initial temperature was 45 °C, heating up to 95 °C, holding for 10 min, cooling down to 25 °C and holding for another 10 min. Both the heating and cooling rates were set as 4.5 °C /min, and the measuring cylinder was rotating at 150 rpm [65,69]. The collected data were analyzed by means of the Data Correlator software (Brabender, Duisburg, Germany). The following parameters of the pasting profile were analyzed: PT—pasting temperature, PV—peak viscosity, PVT—temperature at maximum (peak) viscosity, PVt—time needed to reach PV, MV—minimum viscosity, TV—viscosity at 25 °C, FV—final viscosity, SB—setback (SB = TV − MV), BD—break down (BD = PV − MV) and HPSI—hot paste stability index. The obtained data were used for mathematical modeling of the pasting curves course, allowing a more thorough interpretation of the pasting phenomenon, as proposed by Palabiyik et al. [67]. The following parameters were calculated: V_peak_—peak viscosity, r—the time that gives rise to 50% of peak viscosity and s—starch coefficient.

### 3.6. Thermal Properties

The thermal properties of starch gelatinization were obtained using a differential scanning calorimeter, DSC 204F1 Phoenix (Netzsch, Selb, Germany). DSC measurements were made under a nitrogen atmosphere. An empty aluminum pan was used as the reference sample. The calorimeter was calibrated with a multi-point method (Hg, In, Sn, Bi, Zn and CsCl).

Starch gelatinization and retrogradation was investigated by heating up the water–starch dispersion (3:1) in an aluminum pan within the temperatures range of 25–100 °C, with a heating rate of 10°C/min. Based on the obtained thermograms, the following results were calculated: the onset gelatinization temperature (T_Og_), peak (T_Pg_) and end (T_Eg_), and also gelatinization enthalpy (∆H_g_). Afterwards, the cooling sample was stored at 4 ± 1°C for one day. Retrogradation was measured by reheating the sample pan under the same conditions as for gelatinization. Then, the onset (T_Or_), peak (T_Pr_), end (T_Er_) temperature retrogradation and enthalpy (∆H_r_) were evaluated. The degree of retrogradation was calculated as the ratio of gelatinization and retrogradation enthalpies [69].

The glass transition temperature was determined, according to Kowalski et al. (2019), with temperature modifications by heating the starch in aluminum pans (~15 mg) in two steps. The initial step involved heating up the sample to 120 °C (10 °C/min), then cooling to 0 °C (10 °C/min) and re-heating to 250 °C (10 °C/min). The glass transition temperatures were determined on the basis of the second scan, and the onset temperatures (T_Ogt_), middle (T_MIDgt_), inversion (T_INVgt_) and end (T_Egt_) of the transformation were determined. Additionally, the melting peak was characterized by evaluating the onset temperature (T_Om_), peak (T_Pm_) and conclusion (T_Em_), as well as melting enthalpy (∆H_m_) [88].

### 3.7. WAXS Measurements

The wide-angle X-ray scattering (WAXS) characterization of the starch samples was performed with a PANalytical X’Pert PRO MPD diffractometer (Malvern Panalytical Ltd, Malvern, United Kingdom) equipped with Cu X-ray tube (λ Cu-Kα = 1.5419 Å). The powder pattern of starch was collected in a 3−60° 2θ range in reflection mode.

### 3.8. Statistical Analysis

All analyses were performed in at least duplicate/triplicate, and all data were given on a db basis. The obtained results were subjected to the two-way analysis of variance (ANOVA) using Statistica 10.0 PL. For determining the significance between the means, the Tukey test was used. The significance was declared at *p* < 0.05.

## 4. Conclusions

The physicochemical properties of starch are strongly dependent not only on the genetic determinants of the plant that is the source of the polymer (species/ botanical variety), but also on the environmental conditions under which the plants are grown (temperature, insolation) and the applied agrotechnical treatment (irrigation, fertilization). The conducted research proved that the soilless potato cultivation method (with the controlled conditions of temperature and air humidity, as well as irrigation and fertilization) also influenced the quality of the accumulated starch. Additionally, the effect of the applied hydroponic system (with/without nutrient solution recirculation) was also noted. The obtained results indicate that the hydroponic cultivation of plants improved the accumulation of phosphorus in starch, which was an element desired from the technological point of view. The content of the divalent ions (calcium and magnesium) and potassium in starch extracted from the tubers of plants grown in both NFT systems (GH and FT) was significantly lower than in SO starch. For the sodium ions, an inverse relationship was noted. The effect of the type of hydroponic system (open/closed) on the elemental composition of starch was also demonstrated.

The study also confirmed that the cation content was relatively higher in the PS with higher phosphorus levels. The cultivation method also affected the protein and amylose content of the PS. Our results confirmed that the ratio of AM to PS was a relatively constant characteristic of the botanical variety of the plant, but could be modified by cultivation.

Potatoes grown in soil were able to synthesize granules with the largest average diameter, while for FT, starch was the smallest.

The method of plant cultivation also influenced parameters, such as WBC, SP and S. Over the temperature range studied, the starch from hydroponically grown plants had a smaller increase in WBC than SO. Additionally, the swelling power (at most temperatures) and solubility (at the highest temperature) of SO were higher than the other samples.

Cultivation modifies the values of starch pasting process parameters, such as PV, PVt and PVT. The starch paste from soil cultivation showed very good resistance to elevated temperatures, as evidenced by the low BD values and high HPSI values.

Little variation was found in the values of gelatinization temperatures with respect to the method of cultivation. Significant differences were found for enthalpy and degree of retrogradation. The samples from the soil cultivation method did not differ from the GH sample, while both significantly differed from the FT sample. This indicated, on the one hand, the similarity between traditional and hydroponic cultivation, and, on the other hand, it indicated the significant effect of the recirculation of the nutrient solution on the properties of starch and its behavior during the retrogradation process. In the case of the vitreous transition phenomenon and subsequent melting, no statistically significant differentiation among the samples was found.

According to the literature data, all the analyzed PS samples belong to the crystalline polymorphic forms of B-type, and the starches obtained from the potatoes grown in a greenhouse (PS-GH) are characterized by the highest crystallinity.

All aspects related to the chromatic parameters of the investigated starches confirmed the influence of the type of potato cultivation method applied. The starch samples obtained from the tubers grown in the soil (SO) were darker, grayer with a shade of red and yellow compared to the other samples (FT and GH), as could be seen from the lower values of the L* parameter and the higher values of the a*, b*, C* and WI indices.

## Figures and Tables

**Figure 1 molecules-27-00856-f001:**
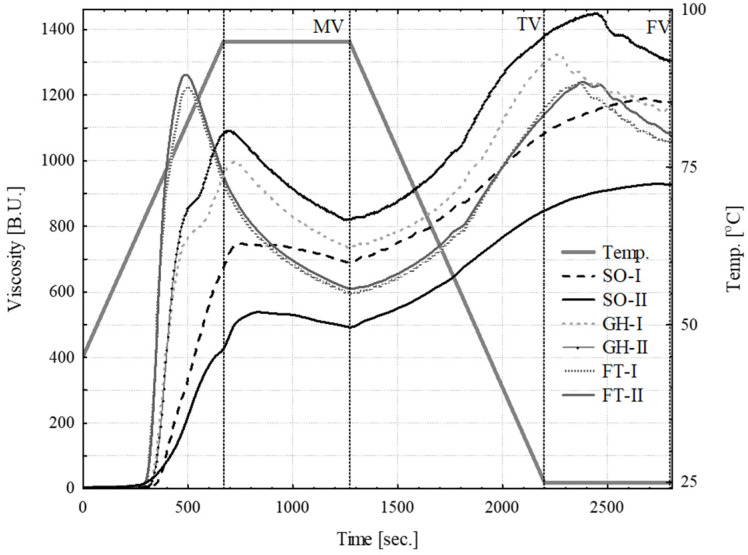
Pasting characteristics of the investigated potato starches (FT—foil tunnel; SO—soil; PT—pasting temperature; MV—minimum viscosity; TV—viscosity at 25 °C; FV—final viscosity).

**Figure 2 molecules-27-00856-f002:**
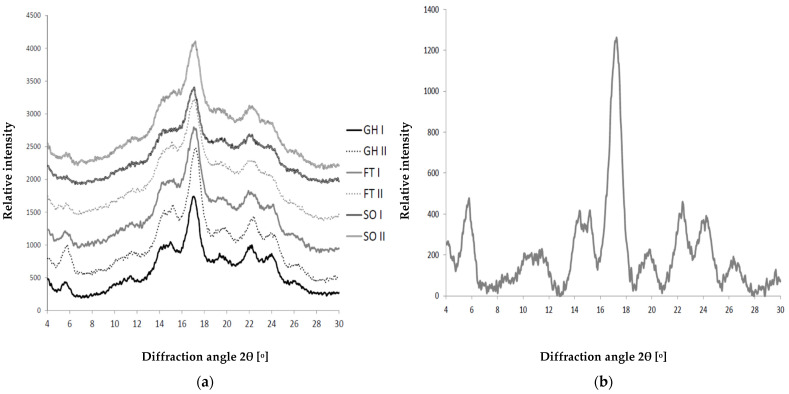
(**a**) XRD patterns for the starches analyzed in the paper. The intensity of the diffraction curves is relative, just to compare the pattern between the samples. (**b**) Crystalline part of the GH II starch diffraction pattern. The intensity is relative. (GH—greenhouse, FT—foil tunnel and SO—soil).

**Table 1 molecules-27-00856-t001:** Chemical composition of potato starches.

Object	Year	Ash (g/100g)	AM (g/100g)	Protein (g/100g)	P	Ca	Mg	Na	K	Molar Fraction			
					(mg/kg)					Ca	Mg	Na	K
GH	I	0.28 ^b^	27.82 ^c^	0.28 ^c^	680.98 ^c^	23.02 ^a^	46.26 ^a^	6.62 ^a^	920.05 ^f^	2.18%	7.24%	1.09%	89.48%
	II	0.27 ^b^	27.81 ^c^	0.30 ^d^	693.01 ^d^	21.89 ^a^	46.36 ^a^	7.21 ^b^	910.87 ^e^	2.10%	7.32%	1.20%	89.38%
FT	I	0.32 ^c^	26.50 ^ab^	0.33 ^e^	720.78 ^e^	27.85 ^b^	48.25 ^b^	9.02 ^d^	892.68 ^c^	2.68%	7.66%	1.51%	88.14%
	II	0.33 ^c^	26.18 ^a^	0.33 ^e^	745.26 ^f^	29.03 ^b^	49.11 ^c^	8.93 ^c^	894.06 ^d^	2.79%	7.77%	1.49%	87.95%
SO	I	0.19 ^a^	27.49 ^c^	0.26 ^a^	602.09 ^b^	96.08 ^c^	68.00 ^d^	21.37 ^e^	605.24 ^b^	11.10%	12.95%	4.30%	71.65%
	II	0.20 ^a^	26.79 ^b^	0.27 ^b^	540.25 ^a^	98.41 ^c^	73.54 ^e^	25.02 ^f^	524.35 ^a^	12.29%	15.14%	5.44%	67.12%
GH		0.28 ^b^	27.82 ^c^	0.29 ^b^	687.00 ^b^	22.46 ^a^	46.31 ^a^	6.92 ^a^	915.46 ^c^	2.14%	7.28%	1.15%	89.43%
FT		0.33 ^c^	26.34 ^a^	0.33 ^c^	733.02 ^c^	28.44 ^b^	48.68 ^b^	8.98 ^b^	893.37 ^b^	2.73%	7.72%	1.50%	88.04%
SO		0.20 ^a^	27.04 ^b^	0.27 ^a^	571.17 ^a^	97.25 ^c^	70.75 ^c^	23.2 ^c^	564.8 ^a^	11.67%	14.00%	4.85%	69.48%
	I	0.27 ^a^	27.27 ^b^	0.29 ^a^	667.95 ^b^	48.99 ^a^	56.00 ^b^	12.34 ^a^	805.99 ^b^	4.95%	9.34%	2.17%	83.54%
	II	0.27 ^a^	26.86 ^a^	0.30 ^b^	659.51 ^a^	49.78 ^b^	54.5 ^a^	13.72 ^b^	776.43 ^a^	5.19%	9.37%	2.49%	82.95%

^a–f^—Values in the same column with different letters in the superscript are significantly different (*p* < 0.05) (*n* = 3); GH—greenhouse, FT—foil tunnel and SO—soil.

**Table 2 molecules-27-00856-t002:** Color parameters and equivalent diameters of the potato starch samples.

Object	Year	Diameters				Color Parameters					ΔE				
		Mean (µm)	<30 µm (%)	30–70 µm (%)	>70 µm (%)	L*	a*	b*	C*	WI	GH-I	GH-II	FT-I	FT-II	SO-I
GH	I	23.5 ^b^	73.8	26.1	0.1	92.01 ^e^	−0.05 ^c^	2.94 ^c^	2.94 ^c^	8.51 ^a^					
	II	22.6 ^a^	82.3	17.6	0.1	92.08 ^e^	−0.16 ^a^	2.72 ^b^	2.73 ^b^	8.38 ^a^	0.3				
FT	I	26.7 ^cd^	64.9	35.0	0.1	91.35 ^d^	−0.09 ^b^	2.28 ^a^	2.29 ^a^	8.94 ^b^	0.9	0.9			
	II	22.1 ^a^	79.5	20.4	0.1	87.36 ^c^	−0.13 ^ab^	2.86 ^bc^	2.86 ^bc^	8.96 ^b^	4.7	4.7	4.0		
SO	I	26.3 ^c^	70.2	29.4	0.4	82.37 ^a^	0.52 ^d^	6.09 ^d^	6.11 ^d^	18.66 ^d^	10.2	10.3	9.8	6.0	
	II	27.4 ^d^	65.4	34.2	0.4	84.86 ^b^	0.58 ^e^	6.03 ^d^	6.06 ^d^	16.31 ^c^	7.8	8.0	7.5	4.1	2.5
GH		22.9 ^a^	78.6	21.3	0.1	92.05 ^c^	−0.11 ^a^	2.83 ^b^	2.84 ^b^	8.45 ^a^					
FT		25.2 ^b^	68.8	31.2	0.1	89.36 ^b^	−0.11 ^a^	2.57 ^a^	2.57 ^a^	8.95 ^b^					
SO		27.1 ^c^	67.0	32.6	0.4	83.62 ^a^	0.55 ^b^	6.06 ^c^	6.09 ^c^	17.48 ^c^					
	I	23.6 ^a^	69.0	30.9	0.1	88.58 ^b^	0.13 ^b^	3.77 ^a^	3.78 ^a^	12.04 ^a^					
	II	25.4 ^b^	77.5	22.3	0.1	88.1 ^a^	0.1 ^a^	3.87 ^b^	3.88 ^b^	12.55 ^b^					

^a–e^—Values in the same column with different letters in the superscript are significantly different (*p* < 0.05) (*n* = 3); GH—greenhouse, FT—foil tunnel, and SO—soil. Color parameters: L*—lightness, a*—color parameter ranging from green, b*—color parameter ranging from blue to yellow (b*), C*—chroma, and WI—whiteness index.

**Table 3 molecules-27-00856-t003:** Water binding capacity (WBC), swelling power (SP) and starch solubility (S) measured for investigated potato starches.

Object	Year	WBC (g/100 g)				SP (%)				S (g/100 g)			
		60 °C	70 °C	80 °C	90 °C	60 °C	70 °C	80 °C	90 °C	60 °C	70 °C	80 °C	90 °C
GH	I	2.39 ^a^	25.57 ^a^	35.50 ^ab^	76.65 ^c^	3.45 ^a^	28.22 ^a^	43.41 ^ab^	100.40 ^b^	1.60 ^b^	5.82 ^ab^	15.91 ^a^	22.66 ^b^
	II	2.55 ^a^	25.84 ^a^	35.68 ^ab^	76.56 ^c^	3.61 ^a^	28.57 ^a^	44.02 ^ab^	100.51 ^b^	1.68 ^b^	6.08 ^bc^	16.66 ^b^	22.84 ^b^
FT	I	2.78 ^b^	26.73 ^b^	31.59 ^a^	71.95 ^b^	3.83 ^b^	29.57 ^c^	39.06 ^a^	93.90 ^a^	1.28 ^a^	6.20 ^c^	16.55 ^b^	22.31 ^a^
	II	2.85 ^b^	26.37 ^b^	33.85 ^ab^	70.21 ^a^	3.90 ^b^	29.04 ^b^	41.74 ^ab^	91.50 ^a^	1.25 ^a^	5.75 ^a^	16.50 ^b^	22.18 ^a^
SO	I	2.83 ^b^	26.74 ^b^	36.4 ^ab^	78.9 ^d^	3.91 ^b^	30.97 ^d^	44.91 ^ab^	105.23 ^c^	1.95 ^c^	10.42 ^d^	16.72 ^b^	24.04 ^c^
	II	3.09 ^c^	27.77 ^c^	37.90 ^b^	78.60 ^d^	4.17 ^c^	32.57 ^e^	47.00 ^b^	105.08 ^c^	1.99 ^c^	11.67 ^e^	17.24 ^c^	24.24 ^d^
GH		2.47 ^a^	25.71 ^a^	35.59 ^ab^	76.61 ^b^	3.53 ^a^	28.40 ^a^	43.71 ^b^	100.46 ^b^	1.64 ^b^	5.95 ^a^	16.29 ^a^	22.75 ^b^
FT		2.82 ^b^	26.55 ^b^	32.72 ^a^	71.08 ^a^	3.87 ^a^	29.30 ^b^	40.40 ^a^	92.70 ^a^	1.27 ^a^	5.98 ^a^	16.53 ^a^	22.25 ^a^
SO		2.96 ^c^	27.25 ^c^	37.15 ^b^	78.77 ^c^	4.04 ^b^	31.77 ^c^	45.95 ^c^	105.15 ^c^	1.97 ^c^	11.04 ^b^	16.98 ^b^	24.14 ^c^
	I	2.67 ^a^	26.35 ^a^	34.50 ^a^	75.84 ^b^	3.73 ^a^	29.58 ^a^	42.46 ^a^	99.84 ^b^	1.61 ^a^	7.48 ^a^	16.39 ^a^	23.00 ^a^
	II	2.83 ^b^	26.66 ^b^	35.81 ^a^	75.12 ^a^	3.89 ^b^	30.06 ^b^	44.25 ^b^	99.03 ^a^	1.64 ^a^	7.83 ^b^	16.80 ^b^	23.09 ^b^

^a–e^—Values in the same column with different letters in the superscript are significantly different (*p* < 0.05) (*n* = 3); GH—greenhouse, FT—foil tunnel and SO—soil.

**Table 4 molecules-27-00856-t004:** Pasting characteristics of the investigated potato starches.

Object	Year	PT (°C)	PVT (°C)	PVt (sec.)	PV (BU)	MV (BU)	TV (BU)	FV (BU)	BD (BU)	SB (BU)	BD% (%)	SB% (%)	HPSI (%)	V_peak_ (BU)	r (s)	s (-)	R^2^ (-)
GH	I	71.3 ^bc^	95.0 ^b^	726 ^b^	1002 ^c^	730 ^c^	1299 ^d^	1141 ^bc^	266 ^b^	561 ^b^	26.5 ^b^	43.1 ^abc^	84.4 ^b^	907.55	411.25	9.89	0.992
	II	72.1 ^c^	95.0 ^b^	708 ^b^	1095 ^c^	819 ^d^	1375 ^e^	1305 ^d^	275 ^b^	551 ^b^	25.1 ^b^	40.1 ^a^	85.8 ^b^	1015.70	427.38	10.99	0.993
FT	I	70.4 ^ab^	81.8 ^a^	498 ^a^	1234 ^d^	597 ^b^	1162 ^c^	1049 ^b^	636 ^c^	559 ^b^	51.5 ^c^	48.1 ^c^	65.2 ^a^	1193.01	363.97	14.44	0.997
	II	70.3 ^ab^	81.4 ^a^	492 ^a^	1273 ^d^	610 ^b^	1141 ^c^	1087 ^bc^	662 ^c^	527 ^b^	52.0 ^c^	46.2 ^bc^	64.6 ^a^	1249.69	365.08	15.86	0.999
SO	I	70.7 ^abc^	95.0 ^b^	750 ^b^	755 ^b^	689 ^bc^	1083 ^b^	1179 ^cd^	66 ^a^	394 ^a^	8.7 ^a^	36.4 ^a^	96.3 ^c^	831.29	533.51	6.53	0.998
	II	69.5 ^a^	94.7 ^b^	828 ^c^	540 ^a^	492 ^a^	848 ^a^	927 ^a^	48 ^a^	354 ^a^	8.8 ^a^	41.8 ^ab^	96.3 ^c^	596.03	549.99	5.64	0.999
GH		71.7 ^b^	95.0 ^b^	716 ^b^	1049 ^b^	774 ^c^	1337 ^c^	1223 ^c^	270 ^b^	543 ^b^	25.8 ^b^	41.6 ^a^	85.1 ^b^	967.02	415.27	10.15	0.993
FT		70.4 ^a^	81.6 ^a^	493 ^a^	1253 ^c^	603 ^b^	1151 ^b^	1068 ^b^	649 ^c^	556 ^b^	51.7 ^c^	47.1 ^b^	64.9 ^a^	1222.59	364.61	15.09	0.998
SO		70.1 ^a^	94.8 ^b^	803 ^c^	612 ^a^	558 ^a^	926 ^a^	1011 ^a^	54 ^a^	367 ^a^	8.8 ^a^	40.0 ^a^	96.3 ^a^	695.20	534.28	6.34	0.999
	I	70.6 ^a^	89.7 ^a^	639 ^a^	969 ^a^	668 ^b^	1201 ^b^	1112 ^a^	374 ^b^	526 ^b^	32.9 ^b^	43.8 ^a^	79.1 ^a^	838.83	390.97	10.81	0.998
	II	70.8 ^a^	90.3 ^b^	675 ^b^	1045 ^b^	640 ^a^	1121 ^a^	1106 ^a^	328 ^a^	477 ^a^	28.6 ^a^	42.7 ^a^	82.2 ^b^	810.36	385.87	12.34	0.999

^a–e^—Values in the same column with different letters in the superscript are significantly different (*p* < 0.05) (*n* = 3); GH—greenhouse; FT—foil tunnel; SO—soil; PT—pasting temperature; PV—peak viscosity; PVT—temperature at maximum (peak) viscosity; PVt—time needed to reach PV; MV—minimum viscosity; TV—viscosity at 25 °C; FV—final viscosity; SB—setback (SB= TV–MV); BD—breakdown (BD = PV−MV); HPSI—hot paste stability index; V_peak_—peak viscosity; r—the time that gives rise to 50% of peak viscosity and s—starch coefficient.

**Table 5 molecules-27-00856-t005:** Thermal properties of the investigated starches.

Object	Year	Gelatinization				Retrogradation					Glass Transition					Melting Peak			
		ΔH_g_ (J/g)	T_Pg_	T_Og_	T_Eg_	ΔH_r_ (J/g)	T_Pr_	T_Or_	T_Er_	R (%)	T_Ogt_	T_MIDgt_	T_INVgt_	T_Egt_	ΔC_p_ (J/g°C)	T_Om_	T_Pm_	T_Em_	ΔH_m_ (J)
			(°C)				(°C)				(°C)					(°C)			
GH	I	16.79 ^a^	74.00 ^b^	69.30 ^b^	79.97 ^a^	8.74 ^a^	62.87 ^a^	44.60 ^a^	80.83 ^a^	52.18 ^a^	81.33 ^a^	93.30 ^a^	91.40^a^	106.10 ^a^	0.08 ^a^	179.30 ^a^	183.30 ^a^	194.07 ^a^	182.97 ^c^
	II	15.59 ^a^	74.20 ^b^	68.20a ^b^	81.30 ^bc^	8.15 ^a^	64.50 ^a^	47.53^bcd^	81.53 ^a^	52.45 ^a^	107.60 ^c^	122.20 ^d^	126.70^d^	143.17 ^d^	0.07 ^a^	190.77 ^a^	194.27 ^a^	203.20 ^b^	131.30 ^a^
FT	I	17.33 ^a^	74.07 ^b^	68.37a ^b^	80.83 ^b^	11.91 ^b^	63.93 ^a^	45.87 ^abc^	76.83 ^a^	68.74 ^bc^	88.77 ^ab^	102.20 ^b^	100.60^ab^	115.63^ab^	0.08 ^a^	186.83 ^a^	190.57 ^a^	199.73^ab^	158.17^abc^
	II	17.04 ^a^	73.50 ^a^	67.93 ^a^	79.83 ^a^	13.92 ^b^	62.73 ^a^	45.40 ^ab^	80.57 ^a^	81.74 ^c^	100.03^bc^	111.30 ^c^	111.90^bc^	122.80^bc^	0.11 ^bc^	180.23 ^a^	184.37 ^a^	194.33 ^a^	159.17^abc^
SO	I	16.98 ^a^	74.90 ^c^	68.87 ^ab^	81.67 ^c^	8.56 ^a^	64.17 ^a^	50.13 ^d^	77.57 ^a^	50.49 ^a^	80.73 ^a^	95.67 ^ab^	95.87^a^	110.30 ^a^	0.12 ^c^	184.67 ^a^	188.30 ^a^	197.20^ab^	139.47 ^ab^
	II	17.20 ^a^	73.53 ^a^	67.87 ^a^	80.03 ^a^	9.30 ^a^	61.83 ^a^	48.53 ^cd^	73.27 ^a^	54.12 ^ab^	100.90^bc^	115.00 ^c^	118.83^cd^	128.90 ^c^	0.10 ^b^	186.70 ^a^	190.70 ^a^	200.50^ab^	132.90 ^ab^
GH		16.19 ^a^	74.1 ^b^	68.75 ^a^	80.63 ^ab^	8.44 ^a^	63.68 ^a^	46.07 ^a^	81.18 ^b^	52.32 ^a^	94.47 ^a^	107.75 ^a^	109.05 ^a^	124.63 ^a^	0.08 ^a^	185.03 ^a^	188.78 ^a^	198.63 ^a^	157.13 ^b^
FT		17.19 ^a^	73.78 ^a^	68.15 ^a^	80.33 ^a^	12.92 ^b^	63.33 ^a^	45.63 ^a^	78.7 ^ab^	75.24 ^b^	94.40 ^a^	106.75 ^a^	106.25 ^a^	119.22 ^a^	0.10 ^b^	183.53 ^a^	187.47 ^a^	197.03 ^a^	158.67 ^b^
SO		17.09 ^a^	74.22 ^b^	68.37 ^a^	80.85 ^b^	8.93 ^a^	63 ^a^	49.33 ^b^	75.42 ^a^	52.28 ^a^	90.82 ^a^	105.33 ^a^	107.35 ^a^	119.60 ^a^	0.11 ^c^	185.68 ^a^	189.5 ^a^	198.85 ^a^	136.18 ^a^
	I	17.03 ^a^	74.32 ^b^	68.84 ^b^	80.82 ^b^	9.74 ^a^	63.66 ^a^	46.87 ^a^	78.41 ^a^	57.14 ^a^	83.61 ^a^	97.06 ^a^	95.96 ^a^	110.68 ^a^	0.10 ^a^	183.6 ^a^	187.39 ^a^	197.00 ^a^	160.20 ^b^
	II	16.61 ^a^	73.74 ^a^	68.00 ^a^	80.39 ^a^	10.45 ^a^	63.02 ^a^	47.16 ^a^	78.46 ^a^	62.76 ^b^	102.84 ^b^	116.17 ^b^	119.14 ^b^	131.62 ^b^	0.10 ^a^	185.9 ^a^	189.78 ^a^	199.34 ^a^	141.12 ^a^

^a–d^—Values in the same column with different letters in the superscript are significantly different (*p* < 0.05) (*n* = 3); GH – greenhouse; FT—foil tunnel; SO—soil; gelatinization: ∆Hg—enthalpy; T_Og_—onset temperature; T_Pg_—peak temperature; T_Eg_—end; retrogradation: ∆Hr—enthalpy; T_Or_—onset temperature; T_Pr_—peak; T_Er_—end temperature; R—degree; glass transition: T_Ogt_—onset temperatures; T_MIDgt_—middle; T_INVgt_—inversion; T_Egt_—end of transformation; melting peak: T_Om_—onset temperature; T_Pm_—peak; T_Em_—conclusion and ∆H_m_—enthalpy.

**Table 6 molecules-27-00856-t006:** Crystallinity of the samples, as calculated from powder diffraction patterns.

Object	Year	Crystallinity [%]
GH	I	40.10
	II	39.95
FT	I	33.27
	II	31.73
SO	I	33.60
	II	34.22

GH—greenhouse, FT—foil tunnel and SO—soil.

## Data Availability

Data will be made available upon request directed to the corresponding author. Proposals will be reviewed and approved by the investigators and collaborators based on scientific merit.

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
