# Peer review of "The Influence of Hydroponic Potato Plant Cultivation on Selected Properties of Starch Isolated from Its Tubers"

_molecules, 2022, doi:10.3390/molecules27030856_

Round 1

Reviewer 1 Report

This paper titled “Influence of hydroponic potato plant cultivation on selected properties of starch isolated from its tubers” is interesting. This manuscript could be considered for publication in Molecules after minor revising. 

My comments are as follow:

  1. Check the abbreviations, the text format of full text, especially figures.  
  2. Please compare your data with previous studies in the result and discussion section.

Author Response

Author's Reply to the Review Report included in the file

Author Response

(The authors gave the same response as above.)

Reviewer 3 Report

The current work is about studying the influence of hydroponic potato cultivation on selected properties of starch. The work is done and presented well. However, writing can be further improved (including typos, like amylase). It would be better if the authors can explain the results better. For instance, how hydroponic cultivation of plants improved the accumulation of phosphorus in starch and why the content of divalent ions (calcium and magnesium) and potassium in starch extracted from tubers of plants grown in both NFT systems (GH and FT) was significantly lower than in SO starch.

Author Response

(The authors gave the same response as above.)
